# Optical Remote Sensing in Provisioning of Ecosystem-Functions Analysis—Review

**DOI:** 10.3390/s23104937

**Published:** 2023-05-20

**Authors:** Pavel Vyvlečka, Vilém Pechanec

**Affiliations:** Department of Geoinformatics, Faculty of Science, Palacky University, 771 46 Olomouc, Czech Republic

**Keywords:** remote sensing, optical sensors, ecosystem function, ecosystem service

## Abstract

Keeping natural ecosystems and their functions in the proper condition is necessary. One of the best contactless monitoring methods is remote sensing, especially optical remote sensing, which is used for vegetation applications. In addition to satellite data, data from ground sensors are necessary for validation or training in ecosystem-function quantification. This article focuses on the ecosystem functions associated with aboveground-biomass production and storage. The study contains an overview of the remote-sensing methods used for ecosystem-function monitoring, especially methods for detecting primary variables linked to ecosystem functions. The related studies are summarized in multiple tables. Most studies use freely available Sentinel-2 or Landsat imagery, with Sentinel-2 mostly producing better results at larger scales and in areas with vegetation. The spatial resolution is a key factor that plays a significant role in the accuracy with which ecosystem functions are quantified. However, factors such as spectral bands, algorithm selection, and validation data are also important. In general, optical data are usable even without supplementary data.

## 1. Introduction

### 1.1. Ecosystem Services and Functions

Natural ecosystems are very complex systems with abiotic and biotic elements. Ecosystem functions are the basic interactions between the components of ecosystems or across ecosystems, and, further, they describe the processes (energy and material transfer) of ecosystems [1]. In the second case, the function is defined by three points [2]:The rate of biological energy or water flow through the ecosystem;The rate of material or nutrient cycling (biogeochemical cycles);Biological or ecological regulation, including both the regulation of organisms by the environment and the regulation of the environment by organisms.

Ecosystem functions are attributes related to the performance of an ecosystem, which are the consequences of one or multiple ecosystem processes [3]. According to another definition, ecosystem functions offer direct and indirect benefits to various species, including humans. In this definition, examples of ecosystem functions include nutrient regulation, food production, or water supply [4]. The essential factors in all ecosystem functions are energy, water, and carbon. These elements are closely interconnected. If a plant does not have sufficient water, the energy flux changes, and the plant changes its evaporation and increases its heat, thus slowing down basic processes such as carbon sequestration [5]. The elements must remain in balance, as they influence each other.

Ecosystem services benefit human well-being, which human populations derive directly or indirectly from ecosystem functions [6]. Services can be described simply as benefits that people receive from ecosystems [7]. Another way to define services is as the specific outcomes of ecosystem functions that directly sustain and enhance human life [8]. The difference between services and functions is defined by the statement that functions can have both inherent and potential anthropocentric values, while services are defined only in terms of their benefits to people [9]. Even though there are known delineations between functions and services, there will always be overlaps across classifications because humans are part of the Earth’s ecosystem. These overlaps are based on basic definitions, since services are based on functions or become services (from functions) if a human evaluates them as beneficial. A typical example is the category “Food and Materials,” which can be considered as both a function and a service. As a function, it provides food and materials for all living and non-living organisms, while as a service, it offers direct benefits to humans, such as the production of firewood, paper made from wood, fruits and berries, or building materials. The results of these services can be monetized. The Economics of Ecosystems and Biodiversity (TEEB) [10] considers ecosystem services from an economic point of view, as the dividends that society receives from natural capital. Maintaining stocks of natural capital thus enables the continued provision of ecosystem-service flows that ensure human well-being [9]. Services create benefits and each service has a different value for each person [11]. However, as already noted, ecosystem services and functions are not strictly separated, and they overlap in some cases (Figure 1).

The awareness of the importance of ecosystem services and functions is beginning to enter the strategic plans of states and international organizations. Some key indicators of ecosystem services or functions are included in the list of Essential Climate Variables (ECVs). The list contains 54 variables, which are divided into 11 groups. There are eight variables in the biosphere group: aboveground biomass, land cover, albedo, fire, land-surface temperature, leaf-area index (LAI), fraction of absorbed photosynthetically active radiation (FAPAR), and soil carbon [12]. In general, this is a group of physical, chemical, and biological variables that critically contribute to the Earth’s climate characteristics and future evolution [4]. The ecosystem structure is also considered one of the classes of EBVs (essential biodiversity variables). This class is formed of the following three variables: live cover fraction, ecosystem distribution, and ecosystem vertical profile [13]. The variables are monitored at the level of individual ecosystems, but their interactions are not.

The classification of ecosystem functions and services has been addressed by several authors [14,15,16,17]. De Groot identified a total of twenty-three ecosystem functions, which were divided into four basic categories: regulation, habitat, production, and information [15]. The issue of the division of ecosystem functions and services has become more widely known thanks to the Millennium Ecosystem Assessment [7]. A similar classification was developed by the TEEB [10]; however, it was primarily concerned with ecosystem services from an economic perspective. Another important development in the field of ecosystem-service classification was the creation of the CICES (Common International Classification of Ecosystem Services) database. Version 4.3 of the database was released in 2013 and replaced by version 5.1 in 2018. The database is divided into three main categories: provisioning, regulating, and cultural. These categories are subdivided further according to biotic and abiotic ecosystems. However, the exact delineation between the classification of ecosystem functions and services is vague. This vagueness lies in the fact that some ecosystem services are classified as classes of ecosystem functions and vice versa. Many authors have failed to accurately delineate the boundary between the classifications of functions and services [6,17,18]. The interdependence between ecosystem functions and human well-being is shown in the ecosystem cascade [19]. In this context, biophysical processes and ecosystem functions are considered as supporting and intermediate services and ecosystem services as final services. Final services are those that can be harvested or gained from the ecosystem. Because the concepts already overlap, we do not refer to processes and functions as supporting and intermediate services. Ecosystem function appears in the second stage of the cascade (Figure 2) and determines the functions of biophysical processes and structures. The result of the whole cascade is the value that every service represents for human well-being.

### 1.2. Data

Due to the advent of new sensor technologies, an increasing number of satellites and better remote-sensing-data availability are now possible for ecosystem-function or service quantification. This includes direct quantification (defined by physical units) or indirect quantification (defined by dimensionless vegetation indices).

Some satellite-mission operators or companions are already producing prepared datasets of ecosystem-function indicators. Typical examples are land-cover or forest-cover datasets, which can reveal the extent of a given habitat. However, some datasets suffer from low spatial resolution. Another problem may be the non-harmonization of classification schemes. Regional and global classification schemes are different due to the requirements of end users and, mainly, the resolution of the applied data. Global datasets are intended for use worldwide, whereas regional datasets are designed only for a given country or selected climatic region. In these cases, the advantage of high- and very-high-resolution satellite data is the possibility they offer of revealing landcover variability even at larger scales. Accuracy, resolution, and coverage have increased through the use of remote sensing rather than mapping with terrestrial methods. The disadvantage of these datasets is the fixed update period. If an up-to-date land cover of an area is needed, it is necessary to use the current imagery data to perform identifications. Global or continental landcover datasets are provided by many organizations. The European Space Agency (ESA) produces a global landcover dataset annually, with a resolution of 10 m. It is primarily derived from Senitnel-2 data. Its accuracy is 77.9 ±1% in Europe and 76.7 ±.5% overall [20]. Examples of freely available land-cover databases are given in Table 1.

In addition to databases on comprehensive land cover, partial habitats are also available. For forest habitats, databases are available for forest height in 2019 at a resolution of 30 m globally [28], forest-cover change over the years globally [29], and for the forest density, dominant forest type, and forest type in Europe at a resolution of 10 m [30]. Another example is the estimation of aboveground biomass or primary production. The first global aboveground-biomass database was GlobBiomass, which was set up in 2010. Most of the validation points are located in Europe (42,003 points), and the database’s resolution is 100 m at the equator. In Europe, the root mean square error is 31.3 t·ha^−1^, which is 40.3% in relative terms [31]. The next global aboveground-biomass database was the ESA Biomass Climate Change Initiative. The current version is version 3, which was released in 2018. However, versions from 2010 and 2017 are also available. The dataset has a resolution of 100 m. The primary data for the creation of the dataset were Sentinel-1 and ALOS-2 PALSAR-2 radar data. The 2018 dataset has a root-mean-square deviation of 167 t·ha^−1^, which is a relative mean deviation of 73% [32]. Furthermore, the MODIS Gross Primary Production (GPP)/Net Primary Production (NPP) dataset is also available. The dataset has a resolution of 1 kilometer [33]. However, it no longer produces new data, as the MODIS sensor has been decommissioned.

In some cases, the custom modeling of the desired identifier is more suitable. Identifier modeling can be customized to achieve higher accuracy with a sufficient amount of ground-truth training and validation data compared with previously created datasets. The number of ground points depends on the predicted variable and the number of predictors. Optical remote-sensing data and derived secondary variables, such as vegetation indices or texture, can be used as inputs. Optical sensors produce data on a vast scale. The main benefit of remote sensing in the process of ecosystem-function analysis is the collection of up-to-date, repeatable, and non-destructive data. Processed data enable discrete and continuous monitoring. These data are always spatially localized and allow the capture of spatial variability as they cover a large area at one moment. The main pitfall of this method is atmospheric influence. It is necessary to use atmospheric corrections on the images derived. Multispectral sensors are mainly used for satellite remote sensing. They scan not only visible areas but also in the near- or middle-infrared electromagnetic spectrum. In some cases, they even scan in the thermal-infrared or panchromatic band. The panchromatic band is characterized by high or very high spatial resolution and a single band. The band is formed of the total light energy in the visible spectrum. Multispectral data have a higher spectral resolution, consisting of multiple spectral bands with broader bandwidth, instead the lower spatial resolution compared to panchromatic data. The panchromatic band is used to improve spatial resolution for multispectral bands. Another option is to use hyperspectral images. Hyperspectral images have the highest spectral resolution (high number and narrow spectral bands) and can best cover the variability in reflectivity. According to Liang [34], data are divided by spatial resolution into four groups:
Low resolution (>1000 m):a.Sentinel-5P, Meteosat MSG, DSCOVER, NOAA-AVHRR;Medium resolution (100–1000 m):a.Sentinel-3, NOAA-VIIRS, TERRA-MODIS;High resolution (5–100 m):a.Sentinel-2, Landsat-8, Landsat-9, SPOT 5;Very high resolution (<5 m):a.Pléiades Neo-3, Pléiades-1, WorldView-4, QuickBird, Ikonos.

As mentioned above, optical sensors produce large amounts of data. If we focus on freely available optical data, the list becomes much smaller. Currently, the best option is to use Landsat or Sentinel satellites. In addition to current images, one can also download image archives. The table below (Table 2) contains the parameters of freely available data. Landsat 7 is not listed in the table, as newer versions are used for current imagery.

## 2. Remote Sensing of Ecosystem Functions in Research

### 2.1. Studies Related to Ecosystem Unctions

This section explores the potential usage of remote sensing for ecosystem functions. The identification or mapping of ecosystem functions from remote-sensing data has been achieved more widely in the last decade. From Web of Science, 42 records were found by using the following combination of keywords: remote sensing and ecosystem function. However, 207 were found by using remote sensing and ecosystem service as keywords (Figure 3). Ecosystem services are included in Figure 3 due to the frequent confusion between services and functions by authors. These search results are up to date as of 1 July 2022. Only a few studies were published before the year of 2014. After this year, the number of published studies increased.

One of the most important studies in this scientific area is the study by Petorelli [4]. According to this study, ecosystem functions have rarely been studied in large areas. Biodiversity has been monitored by assessing structural and compositional characteristics but not functional characteristics. Attempts to monitor ecosystem functions have been conducted in small areas. Most ecosystem assessments do not consider functions due to the insufficient relevant spatial data. In some cases, ecosystems may respond more rapidly to environmental change than structural or compositional attributes and could be among the most sensitive change indicators in global ecosystem monitoring. Another study discussed emerging concepts in coordinated ecosystem monitoring [18], but practical implementation is still lacking. The study summarized information on the distribution of ecosystem functions, relevant indicators that can identify ecosystem functions, and the sensors that can detect these indicators. The sensor overview was divided into two groups: the studies already conducted using these sensors and the potential uses of the sensors in the future. For example, Sentinel-2 is suitable for the estimation of the NDVI (normalized difference vegetation index), LAI (leaf-area index), FAPAR (fraction of absorbed photosynthetically active radiation), or land cover. These identifiers are related to ecosystem functions such as primary productivity, biomass stock, and carbon sequestration. Landsat satellites have the potential to quantify ecosystem functions linked to surface temperature. Sentinel-1 and other radars with similar wavelengths have the potential to quantify functions linked to aboveground biomass and surface moisture [39]. According to Wagner [40], soil moisture is affected by roughness and the presence of vegetation. This influence negatively affects the reflectance values. The soil moisture retrieved from bare soil and maize is feasible with an RMSE of 7%, while other land-cover types were found to have much higher deviations [41]. It is possible to determine soil moisture at small scales, where radar noise is reduced. The Copernicus Global Land Service generates a SSM (surface-soil moisture) product with a 1-km resolution.

In another study that addressed similar issues, the authors focused mainly on remote sensing in agriculture [42]. The study defined variables that are obtainable from remote sensing. According to Nock [43], variables are defined as properties that vary from the individual (plant) to the group (crop) to the community (region). The given characteristics can be typological (crop type), physical (soil-moisture or plant-surface temperature), chemical (leaf nitrogen content), biological (plant phenology), structural (leaf structure), or geometric (plant density). Nock pointed out that none of these properties are directly measurable by remote sensing. The relationship between radiance and properties must be thoroughly modeled. The relationships between the different remote-sensing methods and selected agronomic characteristics were thus investigated. For example, LIDAR data (method) can be used to determine crop height (feature), optical data can be used to determine the Green Area Index or crop yield, and radar data can be used to determine soil moisture. The Green Area Index and crop height can be considered primary variables (directly related to the process of radiance), whereas crop yield is a secondary variable (indirectly available from remote sensing). Primary variables, such as NDVI, LAI, land cover, surface temperature, and vegetation height, are still used to monitor ecosystem functions or services [44,45]. If a secondary variable is investigated, more factors influence the relationship between the radiance and the variable. Another crucial variable in photosynthesis, hydrological processes, and canopy radiation transfer but not listed on the ECV is the clumping index (CI) [46]. The CI is related to LAI and is defined as the ratio of the effective leaf area index (LAIe) to the leaf-area index (LAI) [47].

The values of the CI vary between 0.3 (clumped canopies) and 1 (randomly distributed foliage elements). The clumping index for different vegetation types is classified in the following order: grass, crops and shrubs, and forests (forests are more clumped than grass) [48]. The index can be obtained by direct, allometric, indirect optical, or proxy methods. The study by Fang [46] comprehensively describes the methods and usage of CI. The clumping index can be retrieved from remote sensing by normalized difference hotspot and the darkspot index (NDHD).

In this study, we inferred the primary variables and established which data-collection method using remote sensing is most appropriate. Optical remote sensing can be seen as the most appropriate source and can be used for the widest range of variables. However, the method chosen depends on the needs of the application (Table 3).

### 2.2. Identification of Ecosystem Functions

This study focuses mainly on the following indicators of ecosystem functions: biomass stock, biomass production, carbon sequestration, habitat extent, and habitat quality. The indicators are related to regulation and provisioning (sometimes called production) functions. Regulation functions mitigate fluctuations of natural and anthropogenic origin. Thus, the amount of biomass regulates the runoff [59], erosion [60,61], disturbance [62], and carbon dioxide levels (carbon sequestration occurs) [63], and it regulates temperature [64]. Provisioning functions provide food, materials, or shelter for organisms. In the final table, the provisioning function is used, but the groups of functions are also very closely related. They are linked by the green component of vegetation, which ensures the functioning of ecophysiological processes. The green component is easily detectable by optical sensors. Based on spectral behavior, habitat types and their quality or health can be distinguished with sufficient spatial resolution. It is simple to detect plants whose fitness is not optimal and distinguish them from healthy plants and different plant types, and to accurately determine the spatial boundaries of a habitat. Biomass production represents the quantity of biomass production, and it is expressed in kg × m^−2^ × year^−1^. Carbon sequestration is expressed as the existing carbon reserve in three carbon reservoirs [65]. These reservoirs consist of aboveground, below-ground, and dead biomass. Biomass stock and carbon sequestration are tightly related functions. During photosynthesis, carbon is sequestered in plants, where it is accumulated. According to the Czech National Forest Inventory, the conversion coefficient from aboveground biomass to carbon is 0.49 for deciduous forests and 0.51 for coniferous forests [66]. Aboveground biomass is most often expressed in t·ha^−1^. For quantifying aboveground biomass in a specific habitat, it is necessary to identify the boundaries of the habitat. The indicators were selected according to a study in which only expert values were used [67]. Expert values were transferred to habitat groups by means of the weighted method. In this case, about the focus was son enriching research with studies, options, algorithms, data, or data scaling in diverse areas of interest.

Optical remote-sensing measures reflected radiation and, thanks to different bands, provide radiation values in different spectral bands. Each band is specific to the reflectivity for a certain type of surface. These bands are increasingly entering the modeling of ecosystem functions as prediction data. Some variables can be determined by vegetation indices (primary production, biomass stock, or C-factor). However, more accurate results are obtained through the custom modeling of the selected variable when accurate training and validation data are available. The modeling is performed using modeling algorithms such as RF (random forest) [68], SVM (support vector machine) [69], ANN (artificial neural network) [70], or XGBoost (extreme gradient boosting) [71]. In biomass-estimation modeling, ANN and RF are widely used.

As mentioned above, ecosystem functions or primary variables are not directly measurable from remote sensing. In many cases, functions are detectable based on their relationship with vegetation indices. Many studies used vegetation indices, especially NDVI, to estimate biomass [72,73,74,75,76,77,78]. The results were not based directly on the given vegetation indices, but they were modeled using these indices. Vegetation indices are input data (predictors) for modeling, along with spectral bands. However, other variables, such as the digital elevation model [79,80], vegetation fraction [81,82], land cover [33], and GLCM [80,83], are also used in modeling. These predictors can contribute to improvements in modeling accuracy. However, their usefulness depends on the functions that are modeled by the predictors. For some functions, other predictors, such as bioclimatic variables, soil data, or wind speed, are necessary, and, in some cases, they are dispensable and no longer create improvements. An estimation of primary production using only satellite data was more accurate than an estimation combined with meteorological data [84,85]. Some studies indicate that the modeling of primary production without using a clumping index underestimates the results, especially in tropical regions. Thus, global GPP models without and with a clumping index were compared, and an increase of 5.53 PgC · year^−1^ was found when CI was used. However, the model is more effective in tropical regions (where the LAI is high), with an increase of 4.18 PgC · year^−1^ between 20° N and 20° S [86]. The only study demonstrating the opposite effect of the clumping index uses a different model and approach. In this case, the use of the clumping index resulted in a GPP loss of 12.1 PgC · year^−1^ [87]. However, unlike the previous study, it did not use multiple canopy layers with different proportions of sunlit and shaded leaves, but only a single layer divided into shaded and sunlit leaves. Another study, which investigated the effect of CI on the canopy radiation transfer and photosynthesis rate, demonstrated that the use of global CI datasets overestimated the FAPAR values of the sunlit leaves and underestimated the shaded leaves. Again, a 1 PgC · year^−1^ increase was found when global GPP was compared using CI [88].

When modeling biomass estimation, the input data used, the number of ground measurements, the prediction algorithms, the area of interest, and, thus, the overall accuracy of the model change. The number of ground validation measurements varied significantly in the final table. One study had only eight validation measurements and R^2^ 0.58 [89], while other studies demonstrated the following: twenty-four measurements and R^2^ 0.75 [75], forty-five measurements and R^2^ 0.84 [90], seventy validation measurements and R^2^ 0.57 [91], eighty measurements and R^2^ 0.87 [92], and six hundred and sixty-four measurements and R^2^ 0.41 [79].

The number of validation measurements is certainly important. However, in the results, it was found that the selected classification algorithm, the investigated area, scale, satellite data, and the land-cover type are more influential. For example, RF can work efficiently with large amounts of data. The SVM is less sensitive to noise and the unequal distribution of training areas for different classes. The ANN can be used for many inputs and large datasets. When comparing prediction algorithms, linear regression or multi-linear regression were found to have lower biomass-prediction accuracy [77,91,93,94]. Slightly better were logarithmic or polynomial regressions [77,94]. In contrast, some of the best algorithms for biomass prediction were XGBoost and RF [83,91,95,96]. However, both algorithms are suitable for biomass prediction, with XGBoost and RF achieving 86.9% and 84.4% accuracy, respectively, in one study [83]. The ANN algorithm also performed successfully and, similar to the previous two, it is well suited to biomass modeling [79,92]. Some works used data fusion, which is the integration of multiple data sources to obtain more accurate and higher-quality information. Radar and multispectral data are most commonly used for fusion. Biomass estimation from Sentinel-1 and Sentinel-2 data is a typical example [90,95,97,98,99,100]. Nevertheless, when radar data are used alone, the results are usually much worse than when optical data are used alone. One example of this is the comparison of the accuracy of Sentinel-1 and Sentinel-2 (R^2^ = 0.34 and 0.82, respectively) for biomass prediction [90]. There is also potential in radar sensing, but its suitability for large-scale applications has not yet been confirmed [100].

The validation method is also an important factor. It was found that the choice of method can increase or decrease the quality of the regression model. When using classical 10-fold cross-validation, the coefficient of determination was R^2^ = 0.53, whereas, when using spatial 44-fold cross-validation, R^2^ = 0.14 [101]. Spatial cross-validation splits pixels into homogeneous clusters (44 in this case), and a threshold of a particular distance is set. According to another study, processes and variables in ecology are almost always spatially autocorrelated, and it is appropriate to use spatial cross-validation in such cases [102]. A similar view was advocated in another study that added a temporal, hierarchical, and phylogenetic component to spatial clustering. Predictors that depend on each other are usually used in modeling. If the data set is dependent, it is almost always appropriate to use block cross-validation rather than random cross-validation [103].

Another very important parameter is the spatial resolution of the data. The use of a higher resolution can improve the identification and detection of the spectral behavior of individuals (trees), whereas lower resolutions detect the behavior of groups of individuals (forest stand). Lower resolutions cause many scrambled pixels in the transitions between individual habitats, and their results are more inaccurate. On a small scale, this is not a major problem; however, at a large scale, it is an important parameter. The prediction accuracy can be high (R^2^ = 0.71) even when using medium-resolution data (TERRA-MODIS). However, this was observed in a large study area with only one habitat type [94]. If high-resolution data are used at a smaller scale but with many habitats, the accuracy may be lower [104]. Thus, when using high-resolution data at larger scales, model accuracies can range from 0.8 to 0.9 R^2^ [77,82,83,92].

The indicators and primary variables that were selected based on the functions mentioned above were assigned to classes according to the classification presented by Pettorelli [4]. Two classes were created:The provision of food and materials (materials that can be converted to provide energy and nutrition or for purposes other than food);Supporting habitats (suitable living space for individual species).

For each function, identifiers were assigned, followed by the publications in which the identifiers were mentioned. The next table (Table 4) presents an overview of the methods used in ecosystem-function mapping from remote-sensing data. The table contains several attributes. The ecosystem-function attributes and indicators are listed according to the Pettorelli classification [4]. In addition, the satellite (carrier), the study used, the details, and the formula are included.

## 3. Discussion and Conclusions

The importance of ecosystem functions and services has increased in research and everyday life in the last few years. The realization that ecosystem services and functions need to be studied has led many researchers to work on the problem. However, it is still necessary to work on refining and improving the results of ecosystem-function monitoring. With the arrival of a large number of new satellites and new types of sensor, there are more opportunities to monitor or map ecosystem functions. Different methods are emerging to identify or quantify ecosystem functions using many satellites, different areas of interest, and different ground data. Some algorithms for ecosystem-function determination are not valid for other areas of interest or different sensor types. In our opinion, there is a lack of freely available data with very high resolutions. There are no data available that fulfil the following criteria: very high spatial resolution, near-infrared bands, thermal infrared bands, weekly temporal resolution, and free data availability. Currently, such data may be difficult to obtain, but the use of these data would be a major advance in the monitoring of ecosystem services in future years. Data from Sentinel and Landsat satellites have high potential, but there are limitations in their spatial resolution. In a comparison of freely available optical data between Sentinel-2 and Landsat 9, Sentinel had a better spatial resolution (10, 20, and 60 m) and more near-infrared bands (B5, B6, B7, B8, B8a, and B9). Landsat therefore benefits from the presence of thermal infrared (Bands 10 and 11 with 100 m) and panchromatic bands (15 m). area further image type is radar images, which are mainly investigated in relation to the moisture content of the surface. However, there are also limitations regarding spatial resolution due to the noise caused by surface roughness and the volume of the vegetation. Therefore, they are sometimes used together with optical data as supplementary data.

Satellite data enable further processes for obtaining basic qualitative and quantitative values. With good temporal resolution, they describe the current value of attributes reflecting the spatiotemporal variability of the vegetation-growth phase and fitness. However, the time resolution depends on the cloud cover. Sensors on satellites such as Sentinel-2 (MSI) and Landsat 9 (OLI-2; TIRS-2) offer undeniable advantages in terms of freely available data. Despite their lower spatial resolution, these are still high-quality data sources, which represent progress in the field. Sensors produce high-spatial-resolution data. It is planned to send new Sentinel and Landsat satellites into orbit. The new satellites will feature new sensors, and improved resolution is expected. The new Landsat Next is due to be launched around 2030 and should offer 25 spectral bands, a significant improvement in spectral resolution [35].

In ecosystem services, there is no uniform methodology to solve the problem regarding the partition and classification of ecosystem functions. Many authors attempted to define the boundaries between services and functions. However, the results were ambiguous, and the boundary was identified as uncertain, although some authors defined functions and services clearly [2,3,4,5,6,7,8,9]. This study’s classification of ecosystem functions is based on the Pettorelli classification [4], which is, in my opinion, the best classification of ecosystem functions determined from remote sensing. The review (Table 4) refers to typical studies on the quantification or identification of ecosystem functions that are vegetation-related. The list includes studies with a wide range of data used, areas of interest, vegetation indices, modeling approaches, other supporting data, and ground validation data. The aim was to show the variety of the procedures, the resulting algorithms, the data selection, and the scale of the processing. The findings of the review are as follows:Optical data are useful independently for estimating aboveground biomass from satellite data. Radar data cannot yet be used independently for estimating aboveground biomass and serve as supplementary data for optical data [79,90,95,97,117];Sentinel-2 performs better than Landsat in mapping primary variables and attributes associated with biomass in forest stands [77,91,106,119,120,121];Spatial resolution is crucial for biomass estimation [56,76,91,95];The texture and extent of land-cover classes increase the accuracy of biomass estimation. They are useful only as supplementary data [79,81,83];In general, very-high-resolution satellite data are highly useful for habitat mapping [111,112];The best modeling algorithms are XG-Boost, random forest, and artificial neural network [83,91,92,95,96].

Previous studies mostly agree that the best data for habitat mapping are those very high in spatial resolution and containing at least the NIR band. However, when vegetation indices based on visible radiation and indices based on NIR and SWIR radiation were compared, the results diverged. Some studies reported that indices based on visible radiation were more strongly correlated with biomass than indices based on NIR [75,105]. Another study reported that indices based on visible radiation and NIR were equally important [89]. In contrast, one study reported that the Red Edge and NIR bands were the most important [93]. One study reported that currently, remote-sensing data (optical and radar) are not sufficient to determine biomass above 100 to 150 t × ha^−1^ in forest stands [104]. For primary production, one study reported that ground measurements offer far more accurate results than satellite data [109]. However, it should be mentioned that this study was based on satellite data at a resolution of 100 m. The usage of scale greatly affects the final accuracy of the model, regardless of the data used. As mentioned above, parameters such as the spatial resolution, modeling algorithm, scale, area characteristics, and number and quality of ground measurements always affect the accuracy of the prediction. Another parameter that should receive more attention is the validation method, which also affects the quality of the model. However, the validation method is dependent on the aforementioned parameters. If the spatial resolution or the number of training and validation points is poor, the result is poor in terms of both random and spatial cross-validation. If we focus on the vegetation indices used in the monitoring of ecosystem functions, the most frequently used index is the NDVI. The NDVI estimates of biomass or primary production were the most accurate in six cases. The SR and EVI were also frequently used indices. In the final table, thirteen papers focused on vegetation in temperate, nine in subtropical, and seven in tropical climates.

In conclusion, the use of remote-sensing data is a promising way to obtain information on ecosystem functions. The use of these up-to-date, non-destructible data is a very fast method that can be applied over a large area. Ecosystem functions depend on many factors such as the slope, aspect, precipitation, temperature and other bioclimate variables, soil, wind speed, and ecological variables. However, this study is concerned with determining the indicators of ecosystem functions from remote-sensing data and whether they are applied independently. Some applications still need supplementary meteorological, climatic, or hydrological data. A selection of studies demonstrated that functions can be identified, but there is still a need for data from ground measurements. However, only a small amount of ground data is needed (depending on the size and type of the area) compared to measuring the whole area by using ground-based methods. Complex ground measurements are very expensive and time-consuming. If an entire area were to be measured manually, the time cost would be enormous. Work is still needed to improve the identification and quantification of ecosystem functions, but remote sensing has significant potential.

## Figures and Tables

**Figure 1 sensors-23-04937-f001:**
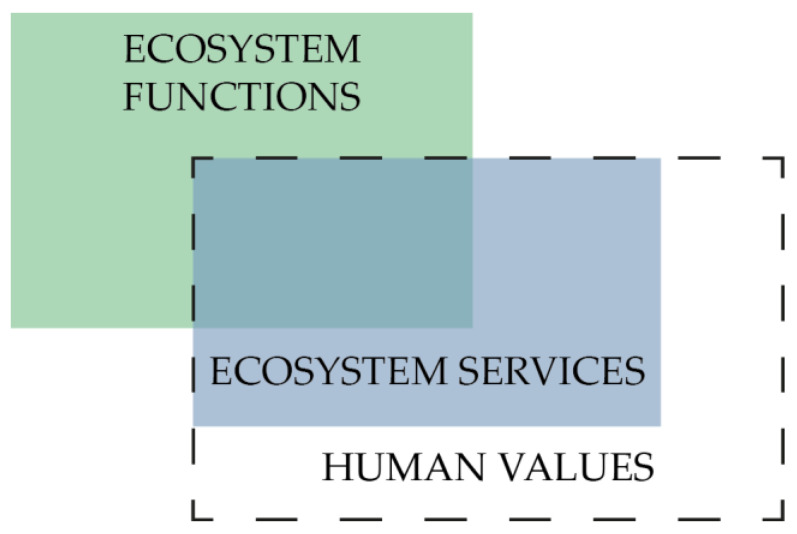
Relationship between functions and services (edited by [4]).

**Figure 2 sensors-23-04937-f002:**
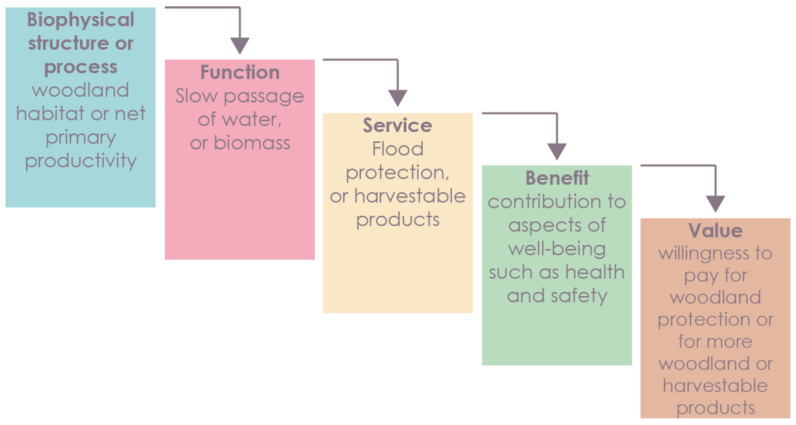
Ecosystem cascade (edited by [19]).

**Figure 3 sensors-23-04937-f003:**
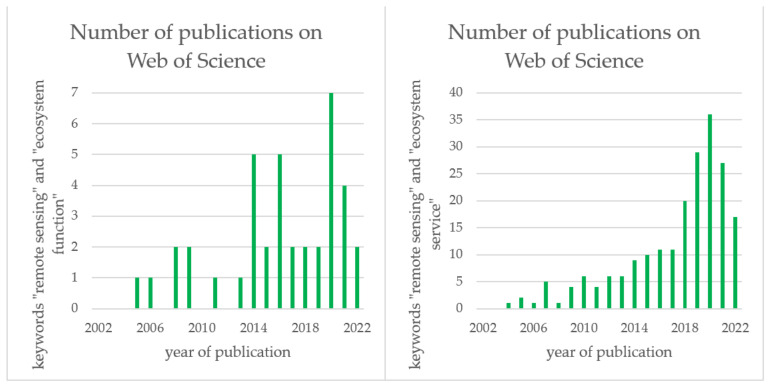
Published articles on Web of Science using the keywords “remote sensing” and “ecosystem function” (**left**) and using the keywords “remote sensing” and “ecosystem service” (**right**).

**Table 1 sensors-23-04937-t001:** Land-cover databases.

Database	Provider	Spatial Resolution	Classes	Latest Version	Update	Coverage	Source
GLC2000	European Environment Agency	250 m	22	2000	Latest version	World	[21]
GLC-SHARE	Food and Agriculture Organization	1000 m	11	2012	Latest version	World	[22]
Corine Land Cover	Copernicus	100 m	44	2018	4 years	Europe	[23]
National Land Cover Database	United States Geological Service	30 m	16	2019	3 years	USA	[24]
Globeland30	National Geomatics Center of China	30 m	10	2020	10 years	World	[25]
ESRI LULC	ESRI	10 m	10	2022	1 year	World	[26]
ESA World Cover	European Space Agency	10 m	11	2021	1 year	World	[27]

**Table 2 sensors-23-04937-t002:** Freely available optical remote-sensing data.

Satellite	Landsat 8	Landsat 9	Sentinel-2	TERRA	NOAA-21 (JPSS-2)
Operator	NASA, USGS	NASA, USGS	ESA	NASA	NOAA
Launch date	11.02.2013	27.09.2021	S2A = 23. 6. 2015; S2B = 7. 3. 2017	18.12.1999	10.11.2022
Orbit	Sun-synchronous at 705 km	Sun-synchronous at 705 km	Sun-synchronous at 786 km	Sun-synchronous at 705 km	Sun-synchronous at 833 km
Orbit Inclination	98.2°	98.2°	98.62°	98.2098°	98.75°
Revisit time	16 days (8 days in combination with Landsat 9)	16 days (8 days in combination with Landsat 8)	10 days for S2A, S2B (5 days together)	1–2 days	16 days
Instruments	OLI; TIRS	OLI-2; TIRS-2	MSI	ASTER	VIIRS
Spatial resolution	15 m and 30 m (OLI); 100 m (TIRS)	15 m and 30 m (OLI-2); 100 m (TIRS-2)	10 m, 20 m and 60 m	15 m, 30 m and 90 m	375 m and 750 m
Radiometric resolution	12-bit (4096 potential values)	14-bit (16,384 potential values)	12-bit (4096 potential values)	8-bit (256) for VNIR, SWIR; 12-bit (4096) for TIR	12-bit (4096 potential values)
Spectral bands	11 bands	11 bands	13 bands	14 bands	22 bands
Source	[35]	[35]	[36]	[37]	[38]

**Table 3 sensors-23-04937-t003:** The most suitable methods for primary variables.

Primary Variables	The Most Suitable Data	Usage Example
Plant Density	multispectral/hyperspectral data	[49]
GAI/LAI	multispectral/hyperspectral data	[50]
Green cover fraction	multispectral/hyperspectral data	[51]
Leaf biochemical content	multispectral/hyperspectral data	[52]
Leaf orientation	Photogrammetry/LIDAR	[53]
Height	Photogrammetry/LIDAR	[54]
FAPAR	multispectral/hyperspectral data	[55]
Albedo	multispectral/hyperspectral data	[56]
Temperature (soil/vegetation)	multispectral/hyperspectral data	[57]
Soil moisture	multispectral/hyperspectral/radar data	[58]

**Table 4 sensors-23-04937-t004:** Overview of methods for ecosystem-function mapping from remote-sensing data.

Type	Ecosystem Function	Indicator	Satellite	Source	Details	Formula (if Available)
Provisioning	Provision of Food and Materials	Biomass stock	GeoEye-1, Pleiades-1A	[89]	Log-biomass estimation in Colombia (tropical climate). The best model in this study used RVI index.	*Log**AGB* [t × ha^−2^] = −3.208 × *RVI* + 2.185
			Landsat 5,7, ALOS PALSAR, LIDAR	[104]	Evaluation of methods used in forest-biomass estimation in Poland, Sweden, Mexico, Africa, and Indonesia.	
			TERRA-MODIS	[96]	Corn-biomass estimation in China (temperate climate). The best algorithm for biomass estimation was XGBoost (R^2^ = 0.78), followed by RF (R^2^ = 0.77). The SVM and ANN were less accurate.	
			Sentinel-1, Sentinel-2	[95]	Forest-aboveground-biomass modeling using random forest in Poland (temperate climate). The model underestimated larger values of biomass (greater than 250 t × ha^−2^) and overestimated small values of biomass (lesser than 100 t × ha^−2^)	
			Landsat, SPOT, RapidEye	[105]	Biomass estimation in eastern Ontario, Canada (temperate climate). Maize biomass correlated best with SRe and soybean biomass correlated best with MTVI2.	*AGB*_*maize*_[g × m^−2^] = 8.247 × *SRe* − 360.98 *AGB*_*soybean*_[g × m^−2^] = 17.26 × *MTVI*2 − 82.339
			Sentinel-2	[93]	Biomass estimation in grasslands in Brazil (subtropical climate). The best results were obtained with NDREI and EVI.	*AGB*[g × m^−2^] = 6.14 + (0.86 × *EVI*) + (3.94 × *NDREI*)
			TERRA– MODIS	[94]	Biomass estimation in grasslands in China (semi-arid continental climate). The best results were obtained with MNDVI.	*AGB*[g × m^−2^] = 51.747 × ln(*MNDVI*) + 155.87
			Sentinel-2	[83]	Biomass estimation in grasslands in China (subtropical climate). The XGBoost models performed better than random forest. Inputs: Sentinel bands, vegetation indices, GLCM (7 × 7 field)	
			Sentinel-2, Landsat	[106]	Comparison between Landsat and Sentinel-2 and bi-temporal (2015, 2019) and uni-temporal (2019) imagery for boreal forest (88% of Picea abies) in Norway (temperate climate). The best results were achieved by Sentinel-2 imagery.	*AGB*[t × ha^−2^] = −185.93 − (485.72 × *B*4/*B*7) + (301 × *B*4/*B*11)
			Sentinel-2	[81]	Biomass modeling in coniferous forest in Ukraine (temperate climate). The best model used NDVI, FCOVER, NDVI, and TVI.	*AGB*[t × ha^−2^] = 716.261 × *NDVI* × *TVI*^−1.644^
			QuickBird	[75]	Forest-biomass estimation for Quercus rotundifolia in Portugal (Mediterranean climate). The best models used NDVI and SR.	*AGB*[t × ha^−2^] = −37.026 + 223.808 × *NDVI* *AGB*[t × ha^−2^] = −47.108 + 42.582 × *SR*
			Sentinel-2, Landsat 8	[77]	Biomass modeling in forests in Pakistan (humid subtropical climate). The best model used NDVI.	*AGB*[t × ha^−2^] = (2099.9 × *NDVI*^2^ − 1635.6 × *NDVI*) + 331.04
			Sentinel-2, Sentinel-1	[90]	Forest-biomass modeling in Indonesia (tropical climate). The best model used VH polarization (Sentinel-1), NDI45, and B6 (Sentinel-2).	*AGB*[t × ha^−2^] = −418.95 + (542.23 × *NDI*45) + (248.48 × *B*6) − (6.56 × *VH*)
			Sentinel-2, Landsat 8, Gaofen-2	[91]	Forest-biomass modeling in China (temperate climate). The best accuracy was modeled using the Gaofen-2 data and random forest algorithm (59.43%). Sentinel-2 had slightly higher accuracy than Landsat-8. Random forest algorithm performed better than multiple linear regression with various remote-sensing data.	
			Landsat, ALOS PALSAR	[79]	Forest-biomass modeling in China (subtropical climate). Landsat provided a more accurate estimation of biomass than ALOS PALSAR. The best algorithm was ANN, whereas RF and kNN were less accurate.	
			Sentinel-2	[92]	Forest-biomass modeling in Western Iran (subtropic climate). The best algorithm to predict biomass was the multi-layer-perceptron artificial neural network (MLPNN), with a coefficient of determination of 0.87. The NDVI and IPVI had the strongest correlation with biomass.	
			Sentinel-2, Landsat 8	[78]	Grass-biomass modeling in Kruger National Park in South Africa (semi-arid subtropical climate). As biomass predictors, Landsat 8 and Sentinel-2 bands were outperformed by NDVI. The accuracy of the NDVI predictor for 2018 was 0.74.	
			Sentinel-1, Sentinel-2	[82]	Biomass estimation in Mediterranean shrublands (subtropic climate). Sentinel-2 had higher accuracy (0,72) than Sentinel-1 (less than 0.6). The fusion model improved accuracy to 0.86. In general, Sentinel-2 showed higher accuracy in shrub-biomass estimation than Sentinel-1, but Sentinel-1 had the potential to indicate vegetation structure.	*AGB*[kg × m^−2^] = (0.148 + 1.735 × NDVI) × sqrt (1 + (2.5σHV-1.2) × FVC/0.5)
			SPOT-5	[76]	Mangrove-forest-biomass estimation in Malaysia (tropical climate). The best model was achieved by using NDVI and GEMI-NDVI.	*AGB*[t × ha^−2^] = 793.676 × NDVI/ 574.770 × GEMI-NDVI-574.219
			Sentinel-2	[80]	Forest--biomass modeling using Sentinel-2 bands and vegetation indices (subtropical monsoon climate).	
		Primary production	Landsat 5,7	[107]	Gross primary production of maize and soybean in Nebraska, USA (temperate climate). The best model for maize used green WDRVI, and the best model for soybean used green NDVI.	*GPP*_*maize*_[gC × m^−2^ × day^−1^] = 2.63 × (*Green* *WDRVI* × *PAR*) − 8.59 *GPP*_*soybean*_[gC × m^−2^ × day^−1^] = 2.86 × (*Green* *NDVI* × *PAR*) − 11.9
			TERRA– MODIS, ENVISAT– MERIS	[84]	Gross primary production in Alpine grasslands in Italy (sub-alpine temperate climate). The MTCI was the best index for estimating GPP (Model 2, direct linear relationship between GPP and MTCI and PAR).	*GPP*[gC × m^−2^ × day^−1^] = *a* × (*MTCI* × *PAR*) + *b*
			TERRA– MODIS	[108]	Gross primary production using vegetation-photosynthesis model and MODIS GPP (tropical climate). Predictions using vegetation indices performed well compared with GPP estimated by eddy covariance.	*GPP*[gC × m^−2^ × day^−1^] = *ε* × α × EVI × PAR
			TERRA– MODIS	[85]	Gross-primary-production modeling in Australia (tropical climate). The best model used EVI.	*GPP*[gC × m^−2^ × day^−1^] = (1.17 × (*EVI*_*MOD*09−0.08_) + 0.03) × *PARTOA*
			TERRA– MODIS; Landsat 5,7	[109]	Gross primary production of maize in China (temperate climate).	*GPP* = *ε* × *APAR* = *ε* × *IPAR* × *FPAR* *IPAR* = *SWRad* − 0.45 *FPAR* = (*NDVI* × 1.24) − 0.168
		Clumping index	TERRA– MODIS	[48]	Estimation of global CI from MODIS BRDF data at 500-m resolution. The model was based on the NDHD index and compared with 33 measurements distributed globally. The resulting accuracy was R^2^ = 0.38.	
			TERRA– MODIS, POLDER	[110]	Determination of global CI from MODIS and POLDER data using NDHD. In total, 72 ground-measurement data were used, and the resulting model accuracy was R2 = 0.8.	
	Supporting Habitats	Habitat extent	Various sources	[111]	Review of habitat mapping depending on the resolution of remote-sensing data.	
			Various sources	[112]	Review of habitat mapping depended on the resolution of remote-sensing data and usage purpose.	
			Landsat	[113]	Estimation of tree-canopy cover (TCC) in Portuguese oak woodlands in Portugal (Mediterranean climate).	*TCC* = 63.626 − 447.22 × *B*5 + 623.837 × *B*4 −714.626 × *B*3 + 281.354 × *B*7
			Sentinel-2, Landsat	[114]	Delineation of habitat types and habitat dynamics in north-west Germany (temperate climate).	
		Habitat quality	ASTER	[115]	Delineation of waterhole bodies (SBR index) and mapping of their condition in Ethiopia and Kenya (tropical climates).	*SBR* = *ASTER*_*Band*3_/*ASTER*_*Band*2_
			AVHRR, MODIS	[116]	Habitat-condition evaluation in Australia (tropical, subtropical climate).	
			Sentinel-2	[117]	Forest-health assessment in the Czech Republic (temperate climate).	
			Sentinel-2	[118]	Potential ecosystem functioning of forests in Dřevnice basin in the Czech Republic (temperate climate).	

## Data Availability

Data or datasets relevant to our paper are available online.

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
