# Peer review of "Optical Remote Sensing in Provisioning of Ecosystem-Functions Analysis—Review"

_sensors, 2023, doi:10.3390/s23104937_

Round 1
Reviewer 1 Report (New Reviewer)
Remote sensing is important for inversing and mapping land surface characteristics related to ecosystem functions. Here Vyvlečka et al., overviewed commonly used optical remote sensing datasets, and their applications in mapping land cover, land surface primary variables, and ecosystem functioning variables. The content is important but further efforts are needed to improve the rigor and comprehension of this paper. I listed my comments as bellows.
Specific comments:
1. Line 8: ‘thus necessary’, remove ‘thus’
2. Line 14-15: revise “the studies are available in a summary table” as “the related studies are summarized as multiple tables”
3. Line 15: ”with Sentinel-2 performing better results” for what kinds of specific studies? Please clarify such conclusion especially in the abstract. I think sentinel-2 not always performance better than landsat.
4. Line 16: I agree spatial resolution could be one factor affecting accuracy but may not necessarily be the primary factor. Please be careful about putting such conclusion in the abstract since it would mislead readers. Perhaps “spatial resolution, inputting variables, algorithm selection, and validation schemes could be primary factors …”
5. Line 27: revise “energy flow” as “energy or water flow”. Energy-water-carbon cycles are most important ecosystem functions. The water flow includes evaporation and plant-mediated transpiration which are critical for adjusting ecosystem functioning. Please see the following papers which give more details about such introduction.
References:
Forzieri, Giovanni, et al. "Increased control of vegetation on global terrestrial energy fluxes." Nature Climate Change 10.4 (2020): 356-362.
Yuan, Kunxiaojia, et al. "Deforestation reshapes land-surface energy-flux partitioning." Environmental Research Letters 16.2 (2021): 024014.
6. Table 1. For the land cover databases, are all the datasets at an annual temporal resolution. An additional column could be added to show the temporal resolution since readers may be interested. Additionally, references are needed for each dataset.
7. Line 174, “Table 1” to “Table 2”
8. Line 233-234, in addition to the listed variables, another variable is also critically important in remote sensing and terrestrial ecosystem functioning but was not listed. The variable is “clumping index”. Please add the “clumping index” in the Table 3 and briefly mention it in the main text. The following papers introduce more details about clumping index and its impacts on global carbon sequestration.
References:
Fang, Hongliang. "Canopy clumping index (CI): A review of methods, characteristics, and applications." Agricultural and Forest Meteorology 303 (2021): 108374.
Li, F., Hao, D., Zhu, Q., Yuan, K., Braghiere, R. K., He, L., ... & Chen, M. (2023). Vegetation clumping modulates global photosynthesis through adjusting canopy light environment. Global Change Biology, 29(3), 731-746.
Braghiere, Rénato Kerches, et al. "Underestimation of global photosynthesis in Earth system models due to representation of vegetation structure." Global Biogeochemical Cycles 33.11 (2019): 1358-1369.
9. Line 248-249, “Thus, the amount of biomass ….temperature”. Citations are needed. For example, why biomass regulates disturbance? Why biomass regulates carbon sequestration. I know one important link between biomass and disturbance is fire, and the relationship between biomass and carbon sequestration is very complex. The following studies are related and give readers more details if they are interested.
Rabin, S. S., Melton, J. R., Lasslop, G., Bachelet, D., Forrest, M., Hantson, S., ... & Arneth, A. (2017). The Fire Modeling Intercomparison Project (FireMIP), phase 1: experimental and analytical protocols with detailed model descriptions. Geoscientific Model Development, 10(3), 1175-1197.
Malhi, Y., Doughty, C. E., Goldsmith, G. R., Metcalfe, D. B., Girardin, C. A., Marthews, T. R., ... & Phillips, O. L. (2015). The linkages between photosynthesis, productivity, growth and biomass in lowland Amazonian forests. Global Change Biology, 21(6), 2283-2295.
10. Line 295-329, in addition to number of validation measurements, algorithms, and spatial resolution, validation method selection is also another very important factor that affect accuracy and estimation of biomass. Please see the following papers:
Ploton, P., Mortier, F., Réjou-Méchain, M., Barbier, N., Picard, N., Rossi, V., ... & Pélissier, R. (2020). Spatial validation reveals poor predictive performance of large-scale ecological mapping models. Nature communications, 11(1), 4540.
Roberts, D. R., Bahn, V., Ciuti, S., Boyce, M. S., Elith, J., Guillera‐Arroita, G., ... & Dormann, C. F. (2017). Cross‐validation strategies for data with temporal, spatial, hierarchical, or phylogenetic structure. Ecography, 40(8), 913-929.
Le Rest, K., Pinaud, D., Monestiez, P., Chadoeuf, J., & Bretagnolle, V. (2014). Spatial leave‐one‐out cross‐validation for variable selection in the presence of spatial autocorrelation. Global ecology and biogeography, 23(7), 811-820.
Author Response
Dear Reviewer 1, Thank you for your substantive comments and suggestions for improving the manuscript. Please find the responses to your comments in the table below. | |
Line 8: ‘thus necessary’, remove ‘thus’ | Accepted |
Line 14-15: revise “the studies are available in a summary table” as “the related studies are summarized as multiple tables” |
Accepted |
Line 15: ”with Sentinel-2 performing better results” for what kinds of specific studies? Please clarify such conclusion especially in the abstract. I think sentinel-2 not always performance better than landsat. |
Sentinel-2 has a higher resolution, which brings a big advantage over Landsat in mapping larger scale areas. Another advantage of Sentinel is the presence of "Red Edge" bands, which provide important information on the condition and type of vegetation. These bands are effective later in the growing season. On the other hand, Landsat contains a thermal band and may produce better results for smaller scales where the resolution is less significant. Of course, we are still talking about vegetation mapping. |
|
Yes, it depends on multiple factors that will affect the accuracy of ecosystem function mapping. However, spatial resolution is important for differentiating between surface types or forest composition. If we consider this in the context of the quality of remote sensing data, spatial resolution is more needed than the number of spectral bands. Of course, in the overall sum, both factors are required. For example, I have a resolution of 10 meters and only three bands (Blue, Green, Red) and a second data with a resolution of 100 meters, but with bands in the infrared and thermal bands. In that case, the results will be more accurate with better spatial resolution. |
Line 27: revise “energy flow” as “energy or water flow”. Energy-water-carbon cycles are most important ecosystem functions. The water flow includes evaporation and plant-mediated transpiration which are critical for adjusting ecosystem functioning. Please see the following papers which give more details about such introduction. |
I agree, energy-water-carbon are essential elements of all ecosystem functions. They are closely related and influence each other. Added to the introduction. |
Table 1. For the land cover databases, are all the datasets at an annual temporal resolution. An additional column could be added to show the temporal resolution since readers may be interested. Additionally, references are needed for each dataset. |
Accepted and edited |
Line 174, “Table 1” to “Table 2” | Edited |
Line 233-234, in addition to the listed variables, another variable is also critically important in remote sensing and terrestrial ecosystem functioning but was not listed. The variable is “clumping index”. Please add the “clumping index” in the Table 3 and briefly mention it in the main text. The following papers introduce more details about clumping index and its impacts on global carbon sequestration. |
Thank you for that very good comment. As you mention, the Clumping Index is also a critical variable that affects other ecosystem functions. The Clumping Index has been added to the overall table and the text to explain it at a basic level. |
Line 248-249, “Thus, the amount of biomass ….temperature”. Citations are needed. For example, why biomass regulates disturbance? Why biomass regulates carbon sequestration. I know one important link between biomass and disturbance is fire, and the relationship between biomass and carbon sequestration is very complex. The following studies are related and give readers more details if they are interested. | Disturbance here meant, for example, a wind disaster or a flood. When higher wind speeds or floods mitigate erosion by trees, mangroves protect coastlines from erosion by large waves in other parts of the world. Nevertheless, it was poorly formulated here, and I added "erosion". I left only fire in the disturbance category, which should be factually correct by now. I have added references to papers that deal with this topic in more depth. |
Line 295-329, in addition to number of validation measurements, algorithms, and spatial resolution, validation method selection is also another very important factor that affect accuracy and estimation of biomass. |
Again, thank you for the appropriate comment. Most biomass modelling models only use random cross-validation, so mentioning a method that is not used and could be very helpful in revealing the "truer" quality of the model is certainly needed. |
Reviewer 2 Report (New Reviewer)
Thank you for this article about the use of remote sensing for ecosystem issues.
Summary
The introduction gives motivations of the document. It describes ecosystem functions and services then technical properties of sensors.
The largest number of articles have been produced in 2020. There Is a relationship between sensor capabilities and the study of ecosystem functions. The document also provides information about carbon sequestration and vegetation as well. It consider different algorithms as RF, SVM, ANN (not to mention energy cost)
The main result is an overview of the methods used in ecosystem functions.
The discussion explain the lack of information on ecosystem services by the ambiguity of the definition. It presents the potential of new articles with improved data quality.
97 references form 1962 to 2022 (30% before 2013, 20% until 2019, 8 are links without timestamp and content can be changed)
Strengths
The text is clear with many references.
Weakness
The reader may expected some studies orientation in conclusion or at least some examples of priorities.
Comments
The introduction does not give a clear organization of the following sections.
What about time series? Do data from different times periods give an accuracy of usages (land cover use) or evolution (increase in activity)?
What about pollution in ecosystem monitoring?
L90: “I would” It is unexpected to think that a multi-author paper use a first-person syntax.
L354: “In my opinion” same comment.
L174: change “table below (Table 1.)” by “Table 2.”
L186: The referred Figure 4. does not exist.
L253: “sensors..” (double dot)
Author Response
Dear Reviewer 2, Thank you for your appropriate and helpful comments on the manuscript. You will find the answers in the table below. | |
The introduction does not give a clear organization of the following sections. |
The introduction is organized to inform the reader about what an ecosystem function is and its position in the ecosystem cascade and what classifications of ecosystem functions/services are now available. The next section then focuses on the data from which ecosystem function indicators can be derived. Primarily these are emerging databases or remote sensing data. The last section also looks at studies already produced on the topic of monitoring ecosystem functions, or the indicators and variables that represent each function. |
What about time series? Do data from different times periods give an accuracy of usages (land cover use) or evolution (increase in activity)? |
Of course, if data from other time periods are used, the results will be different. The study mainly includes data that capture vegetation in the main growing season and does not deal with changes over time. As you mention, the land cover has a specific determination. It is possible to determine the stable parts of the land cover or the changes that have occurred by using images from different time periods. For example, one study monitored biomass in the spring to determine the extent and amount of conifer cover and the other measurement was taken in the main growing season (monitoring deciduous forests). However, the study intends to highlight biomass determination, the data used, and the algorithms and to determine what might be best. |
What about pollution in ecosystem monitoring? | If you mean air pollution monitoring from remote sensing, atmospheric corrections are applied to each image to minimize the effect of the atmosphere and other influences. The corrections are applied to data from ground stations. If you mean ecosystem pollution (e.g., a landfill in a forest), the study does not examine that. |
L90: “I would” It is unexpected to think that a multi-author paper use a first-person syntax. | Accepted and edited |
L354: “In my opinion” same comment. | Accepted and edited |
L174: change “table below (Table 1.)” by “Table 2.” | Accepted and edited |
L186: The referred Figure 4. does not exist. | Accepted and edited |
L253: “sensors..” (double dot) | Accepted and edited |
Round 2
Reviewer 1 Report (New Reviewer)
Thanks for the authors' efforts on addressing my concerns and the current version is okay.
This manuscript is a resubmission of an earlier submission. The following is a list of the peer review reports and author responses from that submission.
Round 1
Reviewer 1 Report
This paper starts way too broadly, including such things as a figure (Figure 3) of the EM spectrum. I would hope that information could be taken as known. Most of the first five pages are super broad summaries that are neither specific enough to be complete or broad enough to be all encompassing. Perhaps a one table summary of publicly available satellite systems would be sufficient. I don't think it is necessary to motivate the reader to understand there is value to remote sensing and how it is used.
Furthermore, the section describing remote sensing (lines 110-137) is incomplete and at times simply incorrect. Spatial resolution is determined by the optical transfer function and ground resolving ability, not solely pixel format. Radiometric resolution is very rarely determined by quantization noise as you describe, but typically by Poisson statistics and system noise. Spectral resolution is given by resolved spectral separation and not just number of channels output. And while the temporal resolution does depend on orbit parameters, that certainly doesn't tell the whole story, as things like frame rate, integration time, and tasking tempo also play a role.
Line 142 should read unmanned aerial vehicle, not unnamed aerial vehicle.
Also, line 184, I would significantly argue against your assertion that the most typical use of LIDAR is to obtain vegetation height. LIDAR is used for quite a lot of things and you need to add qualifiers in there to confine your statement to a narrow set of LIDAR applications.
In line 217, it is unclear what you mean by "negatively affects the reflectance values". Are you implying that soil moisture reduces reflectance? If so, then that very much depends on the spectral bands you are referring to.
In Table 2, line 27, I think you should say thermal sensing and not necessarily multispectral / hyperspectral... as there are not many commercially available thermal hyperspectral sensors.
In line 268, you list that there are no sensors on the market that have a variety of parameters. I disagree with some of your assertions. There are certainly near IR sensors available. And you don't really define what you mean by "very high spatial resolution" in this context. Back in lines 156-164 you list at least five sensors that fall into this category. So, did your definition change?
Table 3 is your capstone data for this paper. Clearly, you have done a lot of literature evaluation to obtain it. However, the sheer variation in parameterizations presented make the case that every data set is unique and that there are an infinite number of ways to fit to remote sensing data that are not related. I was hoping that you would draw trends in the fitting or make some broader claims as to a method to broadly fit differing sets of spectral data. That was however not forthcoming. As it is, you present this long (and likely incomplete) listing of remote sensing investigations of biomass and other ecosystems. Each one looks at different areas with many different sensors, and draws different conclusions with different fitting parameters. The only thing that I took from that is that there is very little overlap in these investigations. Perhaps that is truly your point.
In the end, you conclude that you need to take ground based sensors and go do point measurements anyway.
Overall, I am sure that it took a lot of time and effort to locate, read, and evaluate all the papers listed in table 3. The authors should be commended for their efforts. However, without any attempt to make broader conclusions this paper is somewhat lacking as to how a reader would use it in other situations.
Author Response
Dear reviewer 1, many thanks for all your comments, we try to answer them as best as possible. | |
This paper starts way too broadly, including such things as a figure (Figure 3) of the EM spectrum. I would hope that information could be taken as known. Most of the first five pages are super broad summaries that are neither specific enough to be complete or broad enough to be all encompassing. Perhaps a one table summary of publicly available satellite systems would be sufficient. I don't think it is necessary to motivate the reader to understand there is value to remote sensing and how it is used. |
The introduction has been changed and some parts have been enriched with important information. Overall, an overview of freely available satellite data has been added. |
Furthermore, the section describing remote sensing (lines 110-137) is incomplete and at times simply incorrect. Spatial resolution is determined by the optical transfer function and ground resolving ability, not solely pixel format. Radiometric resolution is very rarely determined by quantization noise as you describe, but typically by Poisson statistics and system noise. Spectral resolution is given by resolved spectral separation and not just number of channels output. And while the temporal resolution does depend on orbit parameters, that certainly doesn't tell the whole story, as things like frame rate, integration time, and tasking tempo also play a role. |
The mentioned section has been modified in the description of remote sensing and resolutions. The new edits should now describe the problem in a better way. |
Line 142 should read unmanned aerial vehicle, not unnamed aerial vehicle. |
Corrected to the right form. The English throughout the article has been revised by MDPI. |
Also, line 184, I would significantly argue against your assertion that the most typical use of LIDAR is to obtain vegetation height. LIDAR is used for quite a lot of things and you need to add qualifiers in there to confine your statement to a narrow set of LIDAR applications. |
Corrected, here it was meant in the context of vegetation mapping. In general, LIDAR has much more usage. |
In line 217, it is unclear what you mean by "negatively affects the reflectance values". Are you implying that soil moisture reduces reflectance? If so, then that very much depends on the spectral bands you are referring to. |
If soil moisture is measured using radar imagery, the reflectance values are then influenced by surface roughness and vegetation density. Thus, the values are then biased. Compared to values that would be measured without an area of vegetation or different surface roughness but with the same surface moisture. This is a typical problem in moisture determination. |
In Table 2, line 27, I think you should say thermal sensing and not necessarily multispectral / hyperspectral... as there are not many commercially available thermal hyperspectral sensors. |
I agree that there are not many thermal hyperspectral satellite sensors. However, I would leave the table with the same statement. There are a number of thermal multispectral sensors (Landsat TM, ASTER, MODIS, AVHRR-3, etc). There are also thermal hyperspectral sensors (AIRS, IASI or CrIS), even if there are not many. |
In line 268, you list that there are no sensors on the market that have a variety of parameters. I disagree with some of your assertions. There are certainly near IR sensors available. And you don't really define what you mean by "very high spatial resolution" in this context. Back in lines 156-164 you list at least five sensors that fall into this category. So, did your definition change? |
Very high resolution data are those that have a spatial resolution better than 5 meters (Liang, 2012). In this case, then, no freely available data meets the condition. There are only high-resolution data (Sentinel-2, Landsat 8, 9). Liang, S.; Li, X.; Wang, J. Advanced Remote Sensing; Academic Press, 2012; ISBN 9780123859549 |
Table 3 is your capstone data for this paper. Clearly, you have done a lot of literature evaluation to obtain it. However, the sheer variation in parameterizations presented make the case that every data set is unique and that there are an infinite number of ways to fit to remote sensing data that are not related. I was hoping that you would draw trends in the fitting or make some broader claims as to a method to broadly fit differing sets of spectral data. That was however not forthcoming. As it is, you present this long (and likely incomplete) listing of remote sensing investigations of biomass and other ecosystems. Each one looks at different areas with many different sensors, and draws different conclusions with different fitting parameters. The only thing that I took from that is that there is very little overlap in these investigations. Perhaps that is truly your point. | The intention was to show the variability of the data, approaches, algorithms, vegetation indices, scales and also the areas of interest. The final text was thus modified according to your comments. |
In the end, you conclude that you need to take ground based sensors and go do point measurements anyway. | In this case, almost all satellite data needs validation data from ground measurements. It is not the whole ground-based mapping, but only a few validation measurements. |
Reviewer 2 Report
Remote sensing has great potential to assess ecosystem functions (the processes that create sink and source products that can be used by other down-gradient ecosystems. And to assess ecosystem services, the sub-set of these sink and source products that benefit human wellbeing. I think you are at the beginning of a great idea, but it is currently not worthy of publication.
Optical remote sensing of ecosystem function analysis - Review
Pavel Vyvlečka 1, * and Vilém Pechanec 1
October 12, 2022
Abstract
1. General thoughts (line 8) about the word ‘disturbance.’ I tend to call myself a disturbance ecologist. The word is often conflated with disturbance that is good (i.e., disturbance regimes) and disturbance that is bad (i.e., anthropogenic disturbance). I try to distinguish between the two using disturbance for the former and perturbation for the latter. But this idea has not taken hold. Just food for thought.
2. “The mentioned problems degrade the ecosystem functions between individual ecosystems and lead to loss of biodiversity, erosion protection, drinking water supplies, the natural home for animals and global warming.” consider. “ These problems degrade ecosystem functions and lead to loss of biodiversity, erosion protection, water quantity and quality, animal habitat, and global warming.”
3. General thoughts (line 11) on the distinction between ecological functions and ecological services. Functions are defined as the physical, chemical, and biological processes following their use in the ecological literature (sensu Odum 1962). While services are defined as the elements of ecosystems that maintain human well-being (Gómez-Baggethun et al. 2010). Again terms are frequently conflated. Your use of drinking water supplies is an example of something that maintains human well-being rather than a process leading to water quality and quantity. And yes, they overlap, but to keep it clear, try to minimize that overlap.
4. It appears that your paper would benefit from edits by a native English speaker. I will not edit the entire document, but there are some grammar and clarity edits that will be necessary. At a minimum, use Grammarly (https://www.grammarly.com/ ) to help with many of the edits.
Introduction
1. The first line (22) is unnecessary and obtuse.
2. Second line is also obtuse, restoration is human intervention.
3. Line 27 services benefit human well-being
4. Lines 30-33 are not services but rather structural elements that support functions. Also, the ECV is a distraction from what I assume your first paragraph is about. You only mention ECV once in the paper.
5. Line 49, I am afraid you may be included in the lack of delineation. Your initial paragraph is confusing. Perhaps because you do spend time defining services, but you need more time defining functions (sensu Odum 1962)
Odum, E. P. 1962. Relationships between structure and function in the ecosystem. Japanese Journal of Ecology 12:108–118.
At this point, I feel that this paper needs a thorough review by an English-speaking editor. I will read the remainder for content only.
1. I am afraid that the entire review of remote sensing is not novel. This is basic information that is not necessary for a peer-reviewed document.
2. I would not say using remote sensing to assess function has been resolved (line 187). There are a lot of unresolved issues both in remote sensing as an assessment tool and assessing function in general!
3. Figure 4 is about functions and services, but the section is about function.
4. Through your “Remote Sensing of Ecosystem Function in Research’” you don’t discuss what functions you are looking at. In the text, you state, “…the ecosystem function or primary variables are not directly measurable from remote sensing. In many cases, functions are detectable based on the relationship with vegetation indices.” This is true in some cases, but patchiness, fragmentation, disturbance regime, aspect, precipitation, continuity, etc., all drive a wide range of ecological processes.
Discussion and Conclusion
Insufficient. There are so many issues, such as scale, resolution, uncertainty, processing time, and signal-to-noise, that list is long.
Author Response
Dear reviewer 2, many thanks for all your comments, we try to answer them as best as possible. | |
General thoughts (line 8) about the word ‘disturbance.’ I tend to call myself a disturbance ecologist. The word is often conflated with disturbance that is good (i.e., disturbance regimes) and disturbance that is bad (i.e., anthropogenic disturbance). I try to distinguish between the two using disturbance for the former and perturbation for the latter. But this idea has not taken hold. Just food for thought. |
Corrected to anthropogenic only because the point of the statement was mainly human activity. |
“The mentioned problems degrade the ecosystem functions between individual ecosystems and lead to loss of biodiversity, erosion protection, drinking water supplies, the natural home for animals and global warming.” consider. “ These problems degrade ecosystem functions and lead to loss of biodiversity, erosion protection, water quantity and quality, animal habitat, and global warming.” |
Changed as recommended. |
General thoughts (line 11) on the distinction between ecological functions and ecological services. Functions are defined as the physical, chemical, and biological processes following their use in the ecological literature (sensu Odum 1962). While services are defined as the elements of ecosystems that maintain human well-being (Gómez-Baggethun et al. 2010). Again terms are frequently conflated. Your use of drinking water supplies is an example of something that maintains human well-being rather than a process leading to water quality and quantity. And yes, they overlap, but to keep it clear, try to minimize that overlap. |
Exactly as you have written, a function provides processes between ecosystems and a service is a function that is a benefit to humans. Drinking water supplies have been changed into water quantity and quality. |
It appears that your paper would benefit from edits by a native English speaker. I will not edit the entire document, but there are some grammar and clarity edits that will be necessary. At a minimum, use Grammarly (https://www.grammarly.com/ ) to help with many of the edits. | The English throughout the article has been revised by MDPI. |
The first line (22) is unnecessary and obtuse | Edited, I agree that the first two sentences were somewhat clumsily written and quite unnecessary. |
Second line is also obtuse, restoration is human intervention. | Corrected. |
Line 27 services benefit human well-being |
After consideration, rewritten in the following form: Ecosystem services benefit human well-being, which human populations derive directly or indirectly from ecosystem functions. |
Lines 30-33 are not services but rather structural elements that support functions. Also, the ECV is a distraction from what I assume your first paragraph is about. You only mention ECV once in the paper. |
Variables are described as key indicators of ecosystem services, not as services themselves, and can thus serve to identify or quantify them. ECVs are thus mentioned in the same way as EBVs that these key indicators are already present in multiple studies and focuses. We are thus talking about indicators that can be identified from remote sensing. |
|
This is just a summary of the fact that many authors have tried to classify ecosystem functions and services, but there has always been uncertainty about the overlapping classes. Classes were then classified vaguely according to the definitions of functions and services. The definitions in the article have been improved. |
I am afraid that the entire review of remote sensing is not novel. This is basic information that is not necessary for a peer-reviewed document. | This is a review of studies in the field of ecosystem function mapping from remote sensing data. This topic is still evolving and needs to be improved. The authors thus bring new approaches in the field, which are summarized in the resulting table. |
I would not say using remote sensing to assess function has been resolved (line 187). There are a lot of unresolved issues both in remote sensing as an assessment tool and assessing function in general! | There was a mistranslation. It is not said that the problem has been solved, but that it has only started to be approached and addressed more in the last decade. |
Figure 4 is about functions and services, but the section is about function | Due to the frequent confusion between the terms service and function by authors of publications and overlapping concepts, the term ecosystem service was also chosen. |
Through your “Remote Sensing of Ecosystem Function in Research’” you don’t discuss what functions you are looking at. In the text, you state, “…the ecosystem function or primary variables are not directly measurable from remote sensing. In many cases, functions are detectable based on the relationship with vegetation indices.” This is true in some cases, but patchiness, fragmentation, disturbance regime, aspect, precipitation, continuity, etc., all drive a wide range of ecological processes. | The focus is mainly on the four ecosystem functions of biomass production, carbon sequestration, climate regulation, and the small water cycle. |
Insufficient. There are so many issues, such as scale, resolution, uncertainty, processing time, and signal-to-noise, that list is long. |
The text has been generally edited according to your comments. English editing has also been included. |
Reviewer 3 Report
This paper attempts to summarize the application of remote sensing in ecosystem function analysis. The paper cited the parameters of ecosystem function analysis and remote sensing data sources. However, the paper did not clearly explain why and how to use remote sensing technology. Therefore, it is necessary to deepen the analysis of the combination of the remote sensing and the ecosystem function.
Major comments:
1. There are too many general concepts introduced in the paper, lacking in in-depth induction.
2. The logical level of the paper is not clear enough and needs to be reorganized.
Minor comments:
1. The abstract lacks the introduction of the research results and needs to be rewritten. It should focus on ecosystem function analysis, optical remote sensing and the combination of the two rather than the introduction of general concepts.
2. The introduction is the accumulation of general concepts, lacking an introduction to why remote sensing technology should be introduced and what problems should be solved. In addition, the research purpose of this paper is not introduced.
3. Line 136-138,these are not relevant to the topic of this section.
4. Spatial resolution is not the only feature of remote sensing. Why give examples of spatial resolution classification? What do you want to say? In addition, Envisat in the table is not the so-called optical remote sensing.
5. There is no MODIS data in recent years. The MODIS in Table 1 now has new alternative data.
6. Section 2, The logical structure is not clear, there is no hierarchy. It is suggested to add subtitles and reorganize the content.
7. In Table 2, the column "The most suitable method" is filled with data instead of method.
8. Line 274,which model?modeling what?
9. Line 277, “This shows both the advantages and disadvantages of remote sensing in ecosystem functions.” How can you get the conclusion? There is no support evidence.
10. Table 3, The contents in the "sensor" column are incorrect. The data in the list are satellite platforms, not sensors.
11.There are many global ecosystem products, such as forest cover products and land use products, etc., which are not mentioned.
12. Papers have no hierarchy and no logical progression,like line 325-337.
13. Section conclusion, the findings from line 368 to 378 are not demonstrated in the text.
14. The concept is not clearly defined. The topic is optical remote sensing. In the paper, radar and laser are also summarized.
Round 2
Reviewer 1 Report
The English reads better in the revision.
I still think the authors could cut out 75% of the information after the initial introduction and prior to section 2 (lines 75 - 238). (reduce to one paragraph and keep table 2?) Focus on the interaction of remote sensing for ecosystem functions and not on defining remote sensing in general. In this context, I would assume the reader would either already understand remote sensing or could investigate it via references.
I am still not particularly happy with the section on sensor resolutions. Apart from overgeneralizing some of the terms (for example - just because a sensor outputs 14-bits of data does not mean there are 16384 useful signal levels... depends on what the noise level is...), I just don't know what it adds to the conclusions of your paper. You can just cite a paper that reviews the various satellite systems without going through the hassle of defining resolution for the reader. Your conclusions are that remote sensing is good... not that here is a new or better way to quantify which remote sensing platforms are better than others.
I do appreciate the effort that went into summarizing all the biomass-relevant remote sensing models. For that, it is worth publishing this article. Some improvement was made in the revision to explain why point ground measurements are not entirely practical. However, there is still minimal attempt to reconcile or generalize the huge variation in models aside from putting a bunch of caveats out there (small sample size, limited coverage, etc.). I would appreciate if the authors could expand on their points of lines 387-392 - but try to dig in to the models a bit more to look for deeper commonality.
Author Response
|
Dear Reviewer 1, thank you for your comments we have tried our best to accommodate and incorporate them. |
I still think the authors could cut out 75% of the information after the initial introduction and prior to section 2 (lines 75 - 238). (reduce to one paragraph and keep table 2?) Focus on the interaction of remote sensing for ecosystem functions and not on defining remote sensing in general. In this context, I would assume the reader would either already understand remote sensing or could investigate it via references. |
Corrected and summarized into a simpler form. Over 50% removed. A simple definition of resolution remained. A simple definition of resolution, optical, radar, lidar RS, and a table of freely available satellites remained. The bullet point with the resolution was retained because we then talk about different satellite systems and the relationship of resolution to the determination of ecosystem functions. This will give the reader who is a layman in remote sensing a quick overview of what resolution means. |
I am still not particularly happy with the section on sensor resolutions. Apart from overgeneralizing some of the terms (for example - just because a sensor outputs 14-bits of data does not mean there are 16384 useful signal levels... depends on what the noise level is...), I just don't know what it adds to the conclusions of your paper. You can just cite a paper that reviews the various satellite systems without going through the hassle of defining resolution for the reader. Your conclusions are that remote sensing is good... not that here is a new or better way to quantify which remote sensing platforms are better than others. |
As I wrote above, the whole section on remote sensing has been edited. However, I have left the small definitions regarding resolution in the text. The 14-bit resolution offers potentially 16384 values. Each spectral band has a SNR (signal-to-noise ratio) that compares the level of the desired signal to the level of background noise. Therefore, realistically, the radiometric resolution will not be completely fulfilled. In Sentinel-2, for example, the bands in the visible part of the spectrum have the best SNR ratio. In the conclusion is stated that when we compare Sentinel-2 and Landsat in mapping primary variables in the forests, Sentinel-2 performs better than the Landsat. |
I do appreciate the effort that went into summarizing all the biomass-relevant remote sensing models. For that, it is worth publishing this article. Some improvement was made in the revision to explain why point ground measurements are not entirely practical. However, there is still minimal attempt to reconcile or generalize the huge variation in models aside from putting a bunch of caveats out there (small sample size, limited coverage, etc.). I would appreciate if the authors could expand on their points of lines 387-392 - but try to dig in to the models a bit more to look for deeper commonality. |
The conclusion has been modified to focus more on drawing out the connections between the models. Every model is unique because of ground measurement. Ground measurement is important part of model creation. But in this case, only a few locations is measured. If the entire site were to be measured manually, the time cost would be extreme. More important is the selection of the algorithm to predict the selected variable. Some studies have agreed that Sentinel-2 is superior to Landsat in mapping forest variables or that radar imagery is not independently useful for estimating aboveground biomass. More findings are described in the discussion and conclusion. |
Reviewer 2 Report
Thoughts – This paper is not ready. This story is not there. There is a lot of literature out there that approaches ecosystem processes using remote sensing. Your table on Line 327 speaks to that. You mention Pettorelli 2018 review, for instance. Also Kerr 2003 Cobello et al. 2012, etc. Google Scholar has 600K papers with those keywords. There are also a lot of papers on ecosystem services and remote sensing, see:
de Araujo Barbosa, C.C., Atkinson, P.M. and Dearing, J.A., 2015. Remote sensing of ecosystem services: A systematic review. Ecological Indicators, 52, pp.430-443.
But your paper struggles with the difference between ecosystem function and ecosystem service. In this second review, it has not cleared up. Maybe you should embrace that and build off of your figure 2—that place between the two. Distinguishing between the two is a mess all over the literature. To the point where it is a big problem with big implications for management, especially with potential tradeoffs of managing toward services. You have something here, and that something might be an attempt to clear up that language and how remote sensing can help once the language is sorted out.
I would be willing to review this again, but ONLY AFTER A MAJOR REWRITE! Not three days later, with a few fixes. This paper needs to find a unique place in the literature and written toward that.
Notes:
Line 45 needs references.
Line 52 You need to decide if this paper is about ecosystem function or ecosystem services (ES). You are trying to force the paper to be about functions, but you keep writing about services. De Groot only writes about services https://scholar.google.com/citations?user=4YZF12kAAAAJ&hl=en&oi=sra I would not turn to him to talk about functions, nor would I turn to Odum to talk about services. If your paper was about measuring services with remote sensing, I think you would solve most of your conflicts.
Line 73 – yes, but the ES should be in the blue box, what is the purpose of the values hatch box? And what do you call the overlap?
line 95. Unnecessary. This is available in any textbook
line 97 as above
lines 75 – 237 are also available in any textbook. This can be summarized in 2 paragraphs!
Line 241 – This is helpful. But I just looked at your paper refed in 31. It states that: The aim of our paper is to present a method for the valuation of selected ecosystem functions (Table 1), demonstrated on the quantification of ecosystem functions in the floodplain. This is very close to my research, and I look forward to reading that paper in detail. But there and here is the conflation between function and service. Table 1 in that paper is similar to the table on line 327. I think the cure for you is to go back to the figure in line 73 and look where functions and services overlap and talk about these four functions/services that exist in this overlap, and that is where you will focus. Although at this point, you should define what these mean. Biomass production is GPP? Carbon Sequestration is NPP? Climate Regulation is microclimate regulation or global regulation.? I don’t have any idea what small water cycle means.
Line 243 – “The functions are selected for the problem addressed, which has not yet included remote sensing data [31].” Nope! Remote sensing-based measures of NPP and GPP - https://www.ntsg.umt.edu/project/modis/mod17.php There are SOOOOO many papers on this. You even state that in your following lines.
Line 327 – table needs a caption. Also, you start with types which are ecosystem services, again to the point that your paper topic is confused. Next, your functions are not those listed on line 241. Sooo I am losing the story thread.
Author Response
Dear Reviewer 2, thank you for your comments we have tried our best to accommodate and incorporate them. | |
Line 45 needs references. | Accepted. |
Line 52 You need to decide if this paper is about ecosystem function or ecosystem services (ES). You are trying to force the paper to be about functions, but you keep writing about services. De Groot only writes about services https://scholar.google.com/citations?user=4YZF12kAAAAJ&hl=en&oi=sra I would not turn to him to talk about functions, nor would I turn to Odum to talk about services. If your paper was about measuring services with remote sensing, I think you would solve most of your conflicts. |
This article is about ecosystem functions. A number of definitions have been added in the introduction to sufficiently describe the difference between functions and services. De Groot discusses the classification of ecosystem functions in [1]. His classification is based on articles [2-4]. In the paper, he discusses the overlap between services and functions. He defines that each function produces services that have a direct or indirect benefit on humans and then assigns them a monetary value. The main idea of using this article is to emphasize that a classification of ecosystem functions has been developed. However, the aforementioned article provides vague information on the difference between function and service; rather, it admits that the two categories overlap. 1. De Groot, R.S.; Wilson, M.A.; Boumans, R.M.J. A Typology for the Classification, Description and Valuation of Ecosystem Functions, Goods and Services. Ecol. Econ. 2002, 41, 393–408, doi:10.1016/S0921-8009(02)00089-7. |
Line 73 – yes, but the ES should be in the blue box, what is the purpose of the values hatch box? And what do you call the overlap? | Ecosystem services have been moved, but the overlap between functions and services remains. The overlap can be well explained in the category "Food and Materials". As an ecosystem function, biomass provides nutrients, materials, or shelter for all organisms. Humans are part of the organism. "Food and Materials" as a service would then specify needs for humans such as firewood, making paper from wood, fruits, or berries. The mentioned services can be converted into monetary value. An ecosystem service will provide a benefit that has a unique value to humans (humans value). |
line 95. Unnecessary. This is available in any textbook | Deleted. |
line 97 as above | Deleted. |
lines 75 – 237 are also available in any textbook. This can be summarized in 2 paragraphs! | Corrected and summarized into a simpler form. Over 50% removed. A simple definition of resolution remained. A simple definition of resolution, optical, radar, lidar RS, and a table of freely available satellites remained. The bullet point with the resolution was retained because we then talk about different satellite systems and the relationship of resolution to the determination of ecosystem functions. This will give the reader who is a layman in remote sensing a quick overview of what resolution means. |
Line 241 – This is helpful. But I just looked at your paper refed in 31. It states that: The aim of our paper is to present a method for the valuation of selected ecosystem functions (Table 1), demonstrated on the quantification of ecosystem functions in the floodplain. This is very close to my research, and I look forward to reading that paper in detail. But there and here is the conflation between function and service. Table 1 in that paper is similar to the table on line 327. I think the cure for you is to go back to the figure in line 73 and look where functions and services overlap and talk about these four functions/services that exist in this overlap, and that is where you will focus. Although at this point, you should define what these mean. Biomass production is GPP? Carbon Sequestration is NPP? Climate Regulation is microclimate regulation or global regulation.? I don’t have any idea what small water cycle means. |
Finally, only three functions were selected from the paper [37]. The selected functions were biomass production, carbon sequestration, and climate regulation. The functions are described in the text. Indicators for these functions were selected for the final table. However, in the table, they are sorted according to the categories of the Pettorelli classification [4].
Biomass production represents quantity of biomass production, and it is expressed in kg × m-2 × year-1. Carbon sequestration is expressed as the existing carbon reserve in three carbon reservoirs. Climate regulation is generally considered in the context of global regulation and is expressed in 1 × m-2 × year-1 units. Climate regulation is a more complex function consisting of several aspects, on of the most important are evapotranspiration and surface temperature.
|
Line 243 – “The functions are selected for the problem addressed, which has not yet included remote sensing data [31].” Nope! Remote sensing-based measures of NPP and GPP - https://www.ntsg.umt.edu/project/modis/mod17.php There are SOOOOO many papers on this. You even state that in your following lines. | Here is a poor clarity of interpretation. The paper in question [37] relied only on expert data that were inferred for the habitat types in the basin. Expert values were transferred to habitat groups by means of the weighted method. Of course, MODIS has a dataset that includes primary production, but the study in question did not use satellite data, and that is the point. |
Line 327 – table needs a caption. Also, you start with types which are ecosystem services, again to the point that your paper topic is confused. Next, your functions are not those listed on line 241. Sooo I am losing the story thread. | The caption has been added. As I wrote before the category "Food and Materials" can be considered both a function and a service. Here it is taken as a function because the indicators "Biomass stock" and "Primary production" are natural processes that have no monetary value and are available to all organisms. The functions in the table are redesigned according to the Pettorelli classification (food and materials, supporting habitats, climate regulation, disturbance regulation, water regulation, barrier effect of vegetation). |